# Bioethanol Production and Alkali Pulp Processes as Sources of Anionic Lignin Surfactants

**DOI:** 10.3390/polym13162703

**Published:** 2021-08-13

**Authors:** Rodrigo Álvarez-Barajas, Antonio A. Cuadri, Francisco J. Navarro, Francisco J. Martínez-Boza, Pedro Partal

**Affiliations:** Pro2TecS-Chemical Process and Product Technology Research Centre, Department of Chemical Engineering, ETSI, Campus de “El Carmen”, Universidad de Huelva, 21071 Huelva, Spain; rodrigo.alvarez@diq.uhu.es (R.Á.-B.); frando@diq.uhu.es (F.J.N.); martinez@diq.uhu.es (F.J.M.-B.); partal@uhu.es (P.P.)

**Keywords:** lignin, emulsion, rheology, microstructure, bitumen, product design

## Abstract

Lignin is an abundant biopolymer with potential value-added applications that depend on biomass source and fractioning method. This work explores the use as emulsifiers of three native lignin-rich product coming from industrial bioethanol production and alkali or Kraft pulping. In addition to their distinctive characteristics, the different molecular organization induced by emulsification pH is expected to interact in various ways at the water-oil interface of the emulsion droplets. Initially, model oil-in-water (O/W) emulsions of a silicone oil will be studied as a function of lignin source, disperse phase concentration and emulsification pH. Once stablished the effect of such variables, emulsion formulations of three potential bitumen rejuvenators (waste vegetable cooking oil, recycled lubricating oil and a 160/220 penetration range soft bitumen). Droplet size distribution, Z-potential and viscous tests conducted on model emulsions have shown that emulsification pH strongly affects stabilization ability of the lignins tested. Regarding bitumen rejuvenators, lignin emulsification capability will be affected by surfactant source, pH and, additionally, by the dispersed phase characteristics. Lower Z-potential values shown by KL at pH 9 and 11 seem to facilitate emulsification of the less polar disperse phases formed by RLUB and bitumen. In any case, lower particle size and higher yield stress values were found for both bioethanol-derived lignins emulsifying RVO and RLUB at pH 13, which are expected to exhibit a longer stability.

## 1. Introduction

Lignin is an abundant renewable polymer that, in many cases, ends up burned to obtain energy as, for instant, lignin contained in black liquor of Kraft process side stream [1,2]. However, potential value-added applications have been described elsewhere, such as antioxidants, antimicrobials or potential building block for chemical modifications, which arise from the high amount of chemicals sites present in the lignin structure [2]. Thus, the complex structure of polymeric lignin has been described elsewhere showing a wide variety of functional groups that depend on its biomass source and fractioning method [3,4]. Particularly, the latter affecting both lignin structural characteristics and industrial potential applications (e.g., surface located carboxylic and phenolic groups, contribute to the biosorption properties of lignin) [3,4,5,6].

Regarding biomass source, hardwood contains approximately 17 y 26% of lignin and is mostly composed of guaiacyl and syringyl units and traces of p-hydroxyphenyl, whereas softwood with around 25 and 31% of lignin mostly contains guaiacyl units, with a small amount of p-hydroxyphenyl units [6,7].

On the other hand, among fractioning methods, lignin derived from alkali or Kraft process is the most extended pulping technology and generates a high amount of residual lignin [8], which is characterized by ionizable groups as phenolic hydroxyl and sulfonic acid added to the lignin structure [9]. Such additional groups result from treatment of lignocellulosic raw material with NaOH and Na_2_S, in an aqueous solution at high temperature [10,11]. As an alternative process, lignin-rich residues may be generated from the bioethanol production process, after biomass saccharification and fermentation [12]. In contrast with Kraft lignin, residual fraction after bioethanol production is composed mostly by lignin but a certain contamination of carbohydrates may be also present, related to non-hydrolysed cellulose during the saccharification [12].

As a result, given its availability as an industrial residue and the biodegradable character of lignin, this polymer is a promising functional polymer that can be involved in the development of novel sustainable and high added-value products. Among them, this work will explore the use of lignin as emulsifier in the manufacture of bituminous emulsions or, more interestingly, of emulsified asphalt rejuvenators. These products, commonly used in road applications, may represent a significant reduction of emissions (e.g., fumes and CO_2_) and energy consumption during asphalt recycling operations, both widely demanded by pavement industry. Thus, the use of reclaimed or aged asphalt pavement (RAP) increases the cost-effectiveness and the sustainability of new asphalt mixes [13]. However, RAP rejuvenation is previously necessary due to the oxidative aging of the binder. Hence, soft virgin bitumen or, more significantly, the use of bio-rejuvenators and rejuvenators derived from waste recycling have been proposed due to their performances and sustainability, since they are recycling products that present a safer usage and, likely, more cost-efficient [13,14,15].

Lignin emulsification ability arises from its structure composed of a hydrophobic backbone and hydrophilic chains, which can allow the interfacial tension between aqueous and oil phases to be reduced. Therefore, lignin can stabilize emulsions by its adsorption in the oil-water interface, causing electrostatic and steric repulsion between oil droplets [16]. To that end, native lignin must be dissolved in water at high pH, being as macromolecules or in a polymeric form with negative net charge, i.e., as anionic surfactants, which are expected to form the so-called anionic bituminous emulsions. 

Cationic emulsions are commonly used in road applications, since their positive droplet surface charge can be more easily destabilized by the most common “acidic” type aggregates (negatively charged), e.g., those with high silica content. Consequently, lignin needs to be modified by amination, turning it into modified cationic modified lignin with protonable groups soluble in acidic media [17]. However, the use of native lignins as anionic emulsifiers of rejuvenators may represent a promising, cheaper and more sustainable alternative for road recycling that involves high rates of reclaimed asphalt (RAP). Under such conditions, surface of RAP aggregates is mostly coated by aged bitumen, which needs to be rejuvenated by emulsified additives. The non-polar character of the aged bitumen may allow either the use of cationic or anionic emulsions [18].

With this aim, this work explores the use of three native lignins as anionic emulsifiers, which are an abundant by-product of industrial processes such as bioethanol production and alkali or kraft pulping. Their particular characteristics derived from their fractioning method, along with their different molecular organization induced by the emulsification pH, are expected to interact in diverse ways with the water-oil interface of the emulsion. Initially model O/W emulsions of silicone oil, with a selected viscosity similar to that of the bitumen during its emulsification, will be studied as a function of lignin source, disperse phase concentration and emulsification pH. Once stablished the effect of such variables, emulsion formulations of three potential bitumen rejuvenators (waste vegetable cooking oil, recycled lubricating oil and a 160/220 penetration range soft bitumen) will be optimized.

## 2. Experimental

### 2.1. Materials

Three commercial lignins derived from Kraft pulping and bioethanol production processes were used in this study. Kraft lignin (KL) was supplied by Sigma Aldrich (Huelva, Spain), with low sulfonate content (4% sulphur) and an average molecular weight, Mw~10,000. Bioethanol derived lignin-reach materials were provided by DONG Energy (Denmark), with two purification degrees. This process uses a hydrothermal pre-treatment to open the cellulosic fibres followed by hydrolysis and fermentation to produce cellulosic ethanol. Depending on processing steps a regular (EthL-1) and a more purified (EthL-2) lignin are produced. As a result, EthL-1 contains 59% lignin, 20 wt.% carbohydrates and 7% proteins, with an average Mw~6800, whereas the most purified, EthL-2, contains 68 wt.% lignin, 19% carbohydrates and 9% proteins, with an average Mw~4600 [19].

Lignin emulsification ability was assessed using oils with different compositions such as silicone oil (FS100 from Esquim S.A., Barcelona, Spain), referred to as SIL, waste vegetable cooking oil (Biolia, Huelva, Spain), referred to as RVO, and a recycled mineral lubricating oil (ECO-350 from Sertego S.L., Madrid, Spain), composed of 66.4% paraffinic, 27.9% naphthenic and 5.7% aromatic compounds, referred to as RLUB. All of them exhibited low viscosity (below 100 mPa·s) at room temperature, which allowed us to easily reproduce, at ambient conditions, bitumen viscosity during its emulsification at high temperature, which is recommended to be lower than 200 mPa·s [20]. Additionally, a bitumen with a penetration grade of 160-220 (Cepsa S.A., Madrid, Spain) was emulsified.

### 2.2. Methods

Emulsification was performed with a homogenizer IKA T25 (Germany) coupled to a S25N–25F SK dispersion tool, at 20,000 rpm for 4 min. All studied oils were emulsified at room temperature, whereas bitumen was blended with the aqueous phase (water and surfactant) to prepare a premixture at 90 °C, prior to emulsification stage. Oil-in-water emulsions with oil concentrations (O%) between 40 and 70 wt.% were studied, while emulsifier concentration was fixed at 0.5 wt.%. Emulsification pH ranged from 9 to 13.

Rheological characterization was carried out by means of steady viscous flow tests at 30 °C. Measurements were conducted in a controlled-stress Haake RS150 (Germany), using a serrated plate-plate geometry PP35 (35 mm diameter, 1 mm gap) to avoid wall-slip phenomena. Steady shear tests were performed in a controlled-strain mode, ranging shear rate from 0.01 and 100 s^−1^. An equilibration time of 120 s guaranteed steady state conditions at every selected shear rate. 

Droplet surface charge was studied by Zeta potential tests, conducted in a Malvern Panalytical Zetasizer (UK). Three measurements were performed for each sample and the average value was reported. Emulsion droplet size distribution (DSD) was determined by laser diffraction in a Malvern Panalytical Mastersizer 2000 (UK). Sauter (D_3,2_) and De-Brouckere (D_4,3_) mean diameters were calculated as follows [21]:(1)$$D_{3,2}=\frac{\sum_{i}n_{i}d_{i}^{3}}{\sum_{i}n_{i}d_{i}^{2}}$$
(2)$$D_{4,3}=\frac{\sum_{i}n_{i}d_{i}^{4}}{\sum_{i}n_{i}d_{i}^{3}}$$
where n_i_ is the number of droplets with diameter d_i_.

## 3. Results and Discussion 

### 3.1. Model Silicone Emulsions: Effect of Disperse Phase Concentration

A first step in the manufacture of any O/W emulsion consists of surfactant dissolution in an aqueous medium that forms the continuous phase of the emulsion. Regarding native lignins, it is well known their solubility in water under alkaline conditions [22]. Thus, all commercial lignin-based emulsifiers were dissolved at pH 13 for this part of the study, which assesses the effect of oil concentration on emulsion microstructure and viscosity. 

Given that all emulsions were formulated with the same surfactant content (0.5 wt.% S), the effective concentration of lignin-based emulsifier in the continuous phase ranged from 0.83 wt.% S (in the emulsion with 40 wt.% O) up to 1.67 wt.% S, for the most concentrated emulsion with 70 wt.% O. As may be seen in Figure 1A, a higher surfactant concentration leads to an increase in the viscosity of the emulsion aqueous phase, more apparent for the most oil-concentrated emulsions. Interestingly, lignin products derived from bioethanol process give rise to more viscous aqueous phases than Kraft lignin. 

In this regard, Estigneev [23] stated the ratio of aliphatic/aromatic hydroxyl groups is the main factor determining the solubility of lignins in alkaline aqueous solutions and, more precisely, the amount of OH groups per aromatic ring. Likewise, Santos et al. [12] pointed out that lignin derived from bioethanol is characterized by a lower phenolic hydroxyl content than Kraft lignin. A higher amount of polar aromatic hydroxyl groups would limit the lignin agglomeration [22,24], and would be related to the lower viscosity exhibited by Kraft lignin solutions at high concentration (Figure 1A). 

All lignin-based surfactants were able to emulsify silicone oil between 40 and 70 wt.% O at pH 13. Most emulsions showed a wide bimodal droplet size distribution (DSD), with a first peak of smaller height located at ca. 4 µm and a second main peak at around 20 µm (Figure 2). 

However, as oil concentration increases the height of first peak decreases in emulsions stabilized by bioethanol-based lignins (Figure 2A) and, eventually, disappears for the most concentrated system stabilized by Kraft lignin (Figure 2B). Furthermore, second peak slightly shifts towards smaller diameters. 

As oil concentration increases, the higher number of droplets along with the narrower DSD are expected to rise droplet-droplet interactions, leading to more viscous systems and with an apparent non-Newtonian character, as confirmed by Figure 3.

When the viscous response is plotted in the form of shear stress-versus-shear rate in log–log scale, a plateau region at intermediate shear rates is observed, which can be related to so-called apparent yield stress. This parameter will be used to study the effect of formulation on emulsion microstructure and stability. The Herschel-Bulkley model describes observed non-Newtonian flow behaviour fairly well:(3)$$\sigma =\sigma_{o}+k\;{\dot{\gamma }}^{n}$$
being σ_o_ the apparent yield stress, k the consistency index and n the flow index. Table 1 gathers the values of Herschel-Bulkley parameters calculated for silicone emulsions prepared with different oil concentrations. 

As may be seen, regardless of the oil concentration or lignin emulsifier considered, all n values are below 1, which evidences the shear-thinning behaviour of these systems. Such a viscous behaviour (characterized by a decrease of emulsion viscosity with shear rate) has been related to droplet deformation, shear-induced deflocculation and/or the non-Newtonian behaviour of the dispersed phase [25,26]. 

Given the range of shear rates studied and the Newtonian behaviour exhibited by the continuous phases shown in Figure 1, the observed shear thinning behaviour should be mainly attributed to a shear-induced deflocculation of the emulsion [27,28]. As expected, an increase in oil concentration leads to more viscous emulsions and with higher yields stress values (Figure 3 and Figure 4B). However, if emulsions with the same concentration of disperse phase are compared, emulsions stabilized with bioethanol-derived emulsifiers show higher yield stress values than emulsions stabilized by KL, despite kraft lignin leads to smaller Sauter (D_3,2_) (up to 65% O) and De-Brouckere (D_4,3_) mean diameters (Figure 4B). 

Initially, such smaller particle sizes should increase emulsion viscosity, due to the increase in specific surface area undergone by its disperse phase [27]. Nevertheless, droplet-droplet flocculation due to entanglements among emulsifier biomolecules seems to control rheological response of these systems. As a result, although KL shows a better emulsification ability at pH 13 (i.e., leading to smaller droplet size), bioethanol-derived lignins are able to form a more developed microstructure of flocculated droplets, which collapses a higher stress values (yield stress) (Figure 4A). Such a flocculated microstructure would be extended forming a three-dimensional network, which would contribute to emulsion stability, as it will discussed later. In this regard, it is worth noting that yield stress results are in good agreement with viscosities measured in Figure 1, suggesting that the microstructure formed by lignins in the continuous phase of the emulsion would play a significant role in its stability and rheology.

In any case, all emulsions formulated at pH 13 exhibit similar Z-potential values around −40 mV (Table 2), confirming the negative charge of the lignin stabilizing droplet interface and the anionic character of the emulsions formed.

### 3.2. Model Silicone Emulsions: Effect of pH

The effect of pH on the viscosity of the emulsion continuous phase, for a selected oil concentration of 65 wt.%, is shown in Figure 1B. Aqueous phases of both lignin-based surfactants derived from bioethanol process showed higher viscosity than Kraft lignin solution at pH between 9 and 11, being more apparent the difference as pH is lower. However, only EthL-1 solution underwent a significant increase in viscosity as pH became less alkaline, whilst viscosity of KL and EthL-2 solutions hardly changed with pH. Observed viscosity changes may be related to different agglomeration behaviours. At high pH values, carboxyl as well as aromatic hydroxyl groups are deprotonated in Kraft lignin, and resultant repulsive Coulomb forces would lead to a better solubility and therefore, to a lower polymer aggregation and viscosity [22].

On the other hand, lining agglomeration is also due to intermolecular hydrogen bonds (formed by aliphatic and aromatic hydroxyl groups) and π–π interactions [22]. Aliphatic hydroxyl groups are able to form stronger hydrogen bonds than aromatic hydroxyls, thus agglomeration behaviour is expected to be more important in bioethanol-derived lignin, characterized by a lower phenolic hydroxyl content than KL [24]. In addition, Maitz et al. [22] pointed out that the π–π stacking effects increase with increasing number of methoxyl groups. When comparing EthL-1 to the more purified EthL-2 [19], this washed sample is expected to be partially demethylated and with a higher amount of polar aromatic hydroxyl groups which, in contrast to methoxyl groups, make more difficult π–π stacking and limit the agglomeration [22,24]. Therefore, its viscosity would be less affected by a pH decrease due to a less extended aggregation, as suggested by Figure 1B. 

As a result, both bioethanol lignins would be more aggregated than KL, particularly at lower pH, as may be deduced from Figure 1. This fact seems to affect their functional properties as emulsifiers. Thus, in contrast with results obtained at pH 13, bioethanol-derived lignins were not capable of stabilizing emulsions with 65 wt.% oil at pH 9 and 11, at least, for one week.

Conversely, KL emulsifier stabilized silicone oil emulsions in the whole pH range tested (Figure 5). Showing monomodal DSD at pH 9 and 11 (Figure 5A) and lower mean particle sizes as medium alkalinity is raised. Specially, D_3,2_ significantly drops at pH 13 (Figure 5B). 

As may be seen in Table 2, all KL emulsions present negative values of Z-potential, higher at pH 13. At this pH, the highest zeta potential of KL stabilized emulsions would result from lignin deprotonated carboxylic, phenolic hydroxyl and acid sulfonic groups [9]. Under conditions less alkaline, in addition to carboxylic groups, sulfonic groups would contribute to the slightly lower zeta potential recorded at pH < 11 (Table 2), which would remain ionized at pH 11 and 9 due to their low pKa ≤ 2 [22,29,30,31]. This fact, in contrast with bioethanol-derived lignins, would allow stable emulsions to be obtained in the whole pH range tested, although emulsion rheology is affected at lower pH. All pH values studied led to shear-thinning behaviours that can be fitted to the Herschel-Bulkley model (Figure 6A), i.e., showing an apparent yield stress and n values below 1. However, emulsion bulk viscosity (and yield stress) at pH 9 significantly decreases if compared with more alkaline pH values (Figure 6B).

### 3.3. Emulsification of Bitumen Rejuvenators

Previous results obtained with O/W silicone model emulsions have pointed out all lignin-based surfactants present good functional properties as emulsifiers at pH 13 in a wide range of oil concentrations. On the contrary, only Kraft lignin extends its emulsification ability towards the lower pHs values of 11 a 9. Subsequently, considering these results, lignins were tested as emulsifiers of bitumen rejuvenators such as a waste vegetable oil (RVO), a recycled lubricant (RLUB) or a soft bitumen (160/220 penetration range). 

As seen with silicone oil, bioethanol-derived lignins exhibited a limited ability of emulsion stabilization, being only able to emulsify waste oils RVO and RLUB only at pH 13 (Figure 7A).

Furthermore, all bitumen emulsions stabilized by these lignins became unstable no matter pH considered. However, Kraft lignin was able to emulsify RVO and RLBU emulsions at pH values between 9 and 13, but the stability of RLBU emulsion was poorer at pH 13 (less than 30 days storage, as seen in Table 2). Likewise, stable bitumen emulsions were obtained only at pH 9 and 11 (Figure 7B, Table 2). All emulsions showed a wide DSD (Figure 7), where main peaks of RVO systems are located at lower diameters (at ca. 5 µm), while bitumen emulsions formulated showed the highest particle sizes. All emulsion presented negative values of Z-potential (negatively charged anionic emulsions) that decreased as pH was less alkaline (Table 2). Initially, a less charged lignin seems to favour emulsification of the less polar rejuvenators, bitumen and RLUB. 

As found for silicone systems, rejuvenator emulsions exhibited a shear thinning behaviour with a trend to reach an apparent yield stress at intermediate shear rates in the log-log curve of shear stress, which has been fitted to the Herschel-Bulkley model (Figure 8).

As may be seen, in contrast with recycled oils RVO and RLUB, pH mainly affects viscosity of bitumen emulsions, being emulsions formulated at pH 9 more viscous than that formulated at pH 11. In any case, RVO and RLUB lead to emulsions with higher viscosity than bitumen. This result is in good agreement with D_3,2_ value displayed in Figure 9A, where RVO and RLUB present the lower values of this mean diameter.

On the other hand, viscous and DSD combined results may be used for selecting best surfactants for every rejuvenator studied (Figure 9). Thus, although most emulsions exhibited a storage stability of at least 30 days, some emulsions formulated with KL and RLUB at pH 13 or with bitumen at pH 9, showed a shorter stability (Table 2). Both systems are characterized by large particle sizes along with low yield stress values (Figure 9), both factors would not prevent droplet motion, and an eventual phase separation. Conversely, lower particle size and higher yield stress values were found for both bioethanol-derived lignins emulsifying RVO and RLUB at pH 13, which are expected to exhibit enhanced stability. Figure 10 seems to confirm this assumption, since these systems hardly change their particle size with time. Regarding bitumen emulsions, best results are achieved with KL at pH 11, although lignin emulsions are characterized by high D_3,2_ and low σ_o_ values. Among all system studied, this emulsion undergoes the largest increase in particle size along storage time (Figure 10).

## 4. Concluding Remarks

Three native lignins have been assessed as emulsifiers of oil-in-water emulsion of three potential bitumen rejuvenators (waste vegetable cooking oil, recycled lubricating oil and 160/220 penetration range soft bitumen). Selected lignins are abundant by-products of industrial processes such as bioethanol production and alkali or Kraft pulping. Previously, lignin emulsification ability has been studied in model silicone emulsions as a function of disperse phase concentration and emulsification pH.

All lignin-based surfactants were able to emulsify silicone oil between 40 and 70 wt.% O at pH 13, giving rise to anionic emulsions with Z-potential values around −40 mV. Most resultant emulsions showed a wide bimodal droplet size distribution and a non-Newtonian viscous character, with shear thinning behaviour and an apparent yield stress. Thus, the observed viscous behaviour has been fitted to the Herschel-Bulkley model. Droplet-droplet flocculation due to entanglements among emulsifier biomolecules seems to control rheological response of these systems. Such an extended flocculated microstructure would contribute to emulsion stability. 

Differences among lignins appear when a less alkaline pH is used during emulsification. Bioethanol-derived lignins are not able to stabilize emulsions with 65 wt.% oil at pH 9 and 11, for at least one week. Conversely, KL emulsifier stabilized silicone oil emulsions in the whole pH range tested. In this regard, different agglomeration behaviours have been observed depending on the fractioning method, where bioethanol-derived lignins would tend to aggregate more than Kraft lignin, particularly at lower pH. The expected higher phenolic hydroxyl content in KL, along with the presence of other deprotonated carboxylic and sulfonic groups, would prevent lignin aggregation and extend surfactant functional properties over a wider pH range, from 9 to 13. 

Similarly, lignin emulsification ability of bitumen rejuvenators depends on surfactant source and pH, but now characteristics of the dispersed phase must be also taken into consideration. Bioethanol-derived lignins exhibited a limited ability of emulsion stabilization, being only capable of emulsifying waste oils RVO and RLUB at pH 13. On the contrary Kraft lignin was capable of emulsifying RVO and RLBU emulsions at pH values between 9 and 13. Likewise, stable bitumen emulsions were obtained only at pH 9 and 11. In this regard, the lower Z-potential values shown by KL at pH 9 and 11 seem to facilitate emulsification of the less polar disperse phases formed by RLUB and bitumen.

In any case, lower particle size and higher yield stress values were found for both bioethanol-derived lignins emulsifying RVO and RLUB at pH 13, which are expected to exhibit a longer stability.

Finally, although this work focuses its application on the important road paving industry, other possible applications could add value to this waste biopolymer as anionic surfactant for cleaning, food, personal care, cosmetic, paint, biomedical, chemical, agricultural and pharmaceutical industry, among others [29]. In this regard, it is worth noting that 100 million tons of lignin are produced annually (including bioethanol, Kraft, sulphite, soda, organosolv processes, among others), but less than 2% is used in commercial products with higher added value [6,29].

## Figures and Tables

**Figure 1 polymers-13-02703-f001:**
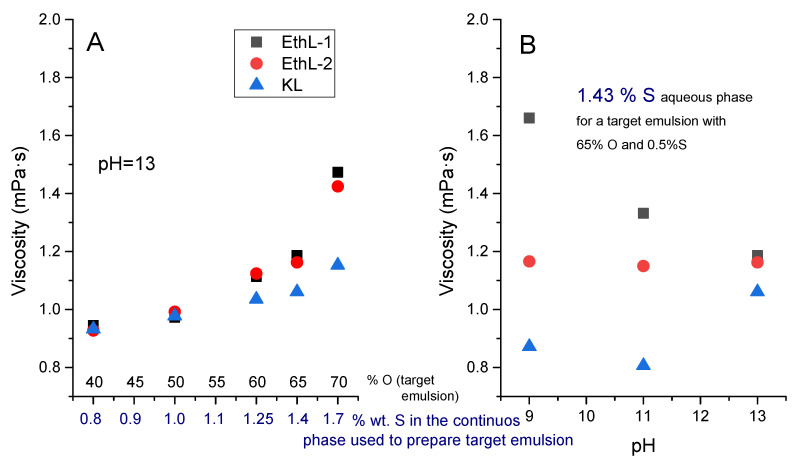
Viscosity of the emulsion continuous phase as a function of: (**A**) selected oil concentration of emulsion prepared at pH = 13; (**B**) emulsification pH for emulsions containing 65% O.

**Figure 2 polymers-13-02703-f002:**
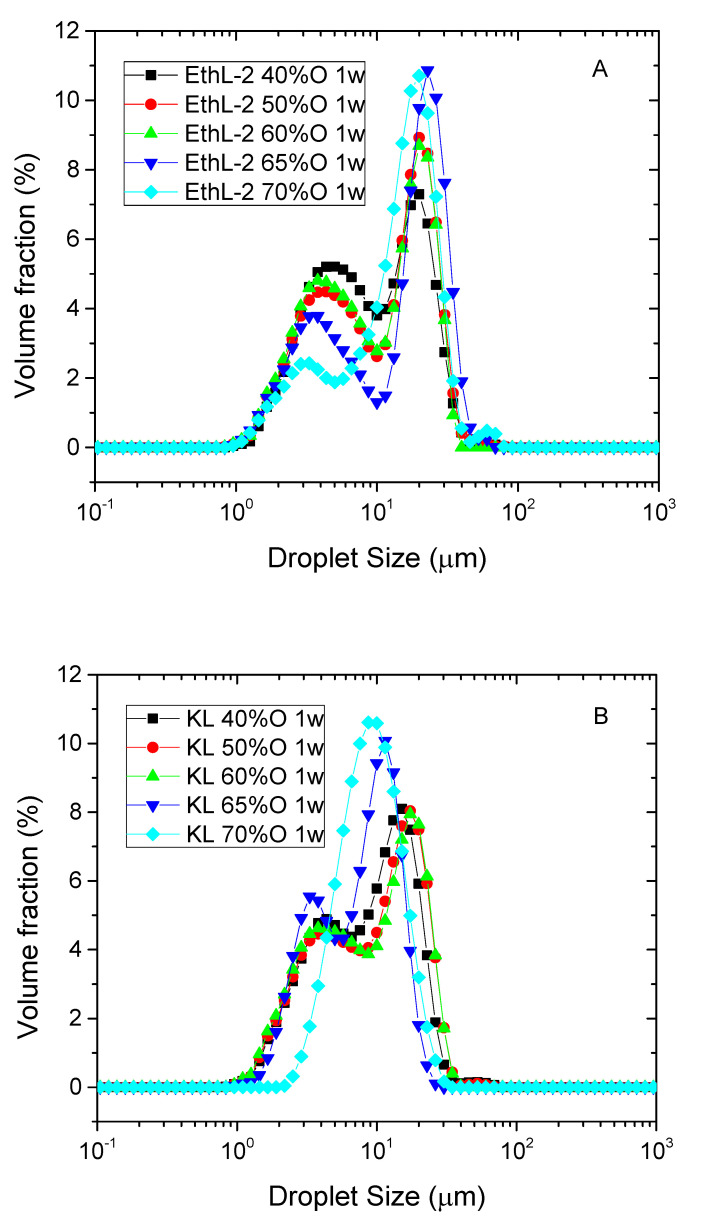
Effect of silicone oil concentration on droplet size distribution of emulsions stabilized by 0.5 wt.% bioethanol (**A**) or Kraft lignin (**B**) at pH = 13, after one week storage.

**Figure 3 polymers-13-02703-f003:**
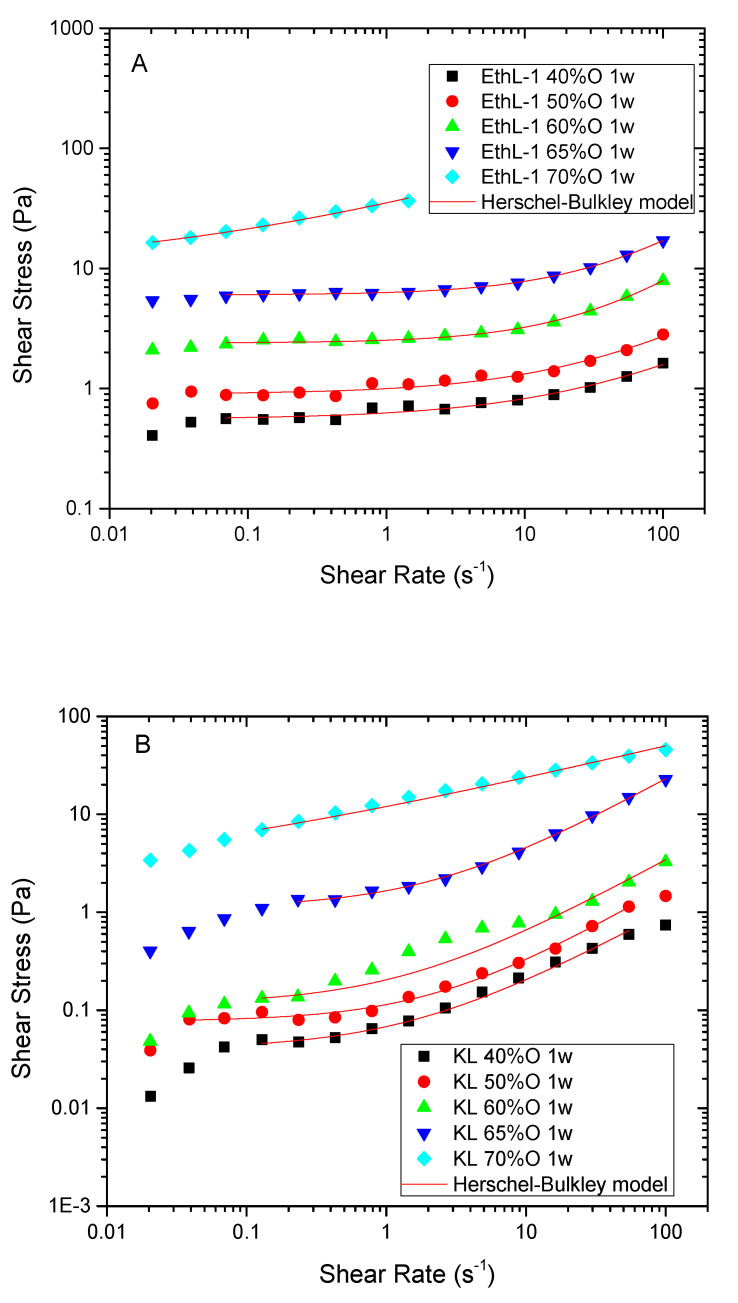
Effect of silicone oil concentration on viscous behaviour of emulsions stabilized by 0.5 wt.% bioethanol (**A**) or Kraft lignin (**B**) at pH = 13, after one week storage.

**Figure 4 polymers-13-02703-f004:**
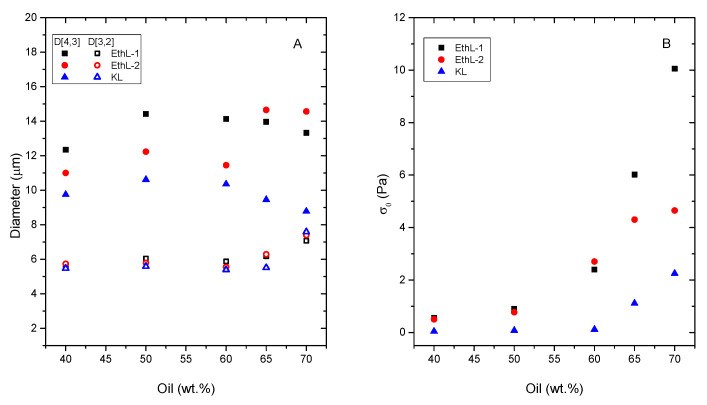
Effect of silicone oil concentration on emulsions parameters: (**A**) Sauter, D_3,2_ and volumetric, D_4,3_, mean diameters; and (**B**) yield stress values determined from Herschel-Bulkley model. All emulsions were emulsified at pH 13, stabilized by 0.5 wt.% lignin and stored for 1 week.

**Figure 5 polymers-13-02703-f005:**
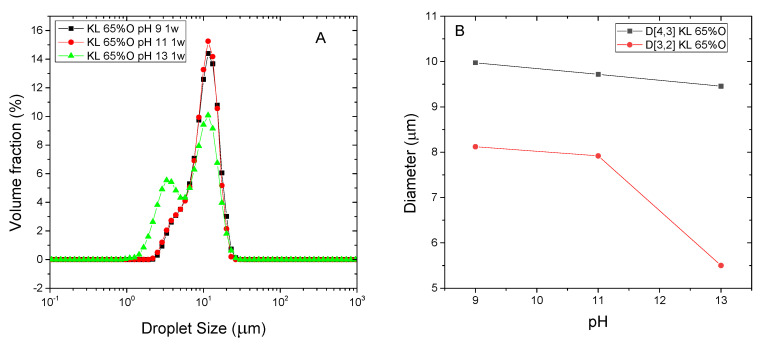
Droplet size distribution (**A**) and characteristic mean diameters (**B**) of silicone oil emulsions stabilized by Kraft lignin as function of pH. All emulsions were formulated with 65 wt.% O, stabilized by 0.5 wt.% lignin and measured after 1 week storage.

**Figure 6 polymers-13-02703-f006:**
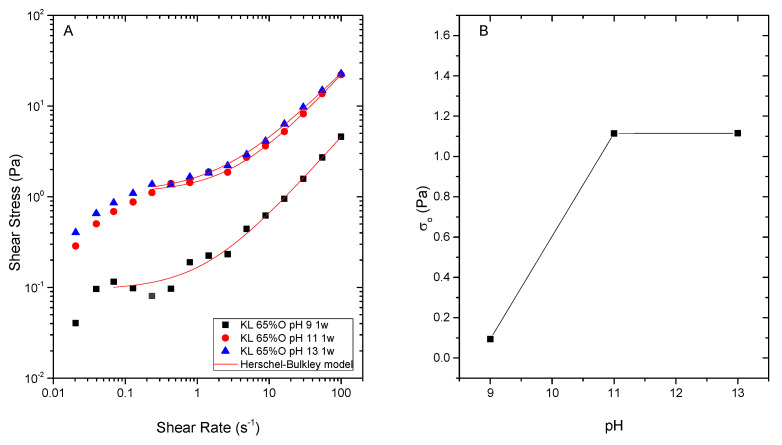
Effect of emulsification pH on silicone emulsions flow behaviour (**A**) and yield stress values (**B**) determined from Herschel-Bulkley model All emulsions were formulated with 65 wt.% O, stabilized by 0.5 wt.% Kraft lignin and measured after 1 week storage.

**Figure 7 polymers-13-02703-f007:**
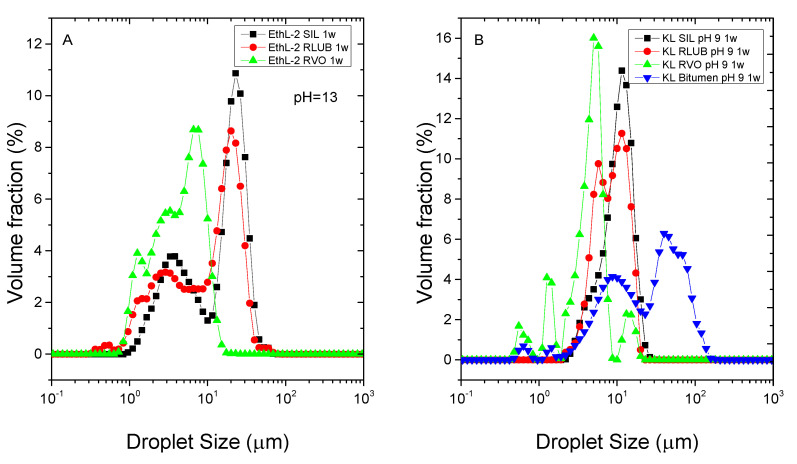
Droplet size distribution of bitumen rejuvenator emulsions stabilized by 0.5 wt.% bioethanol-derived (**A**) and Kraft (**B**) lignins.

**Figure 8 polymers-13-02703-f008:**
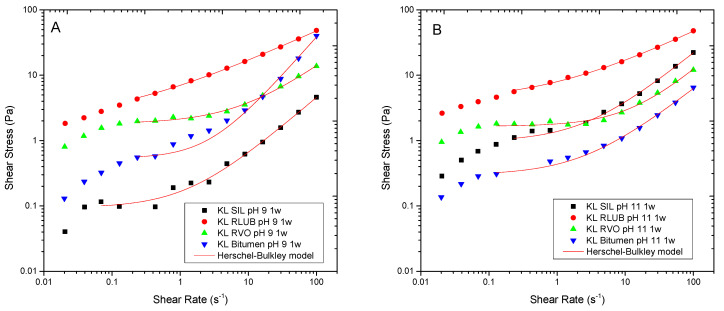
Effect of emulsification pH on the flow behaviour of bitumen emulsions stabilized by Kraft lignin after 1 week storage at pH 9 (**A**) and 11 (**B**).

**Figure 9 polymers-13-02703-f009:**
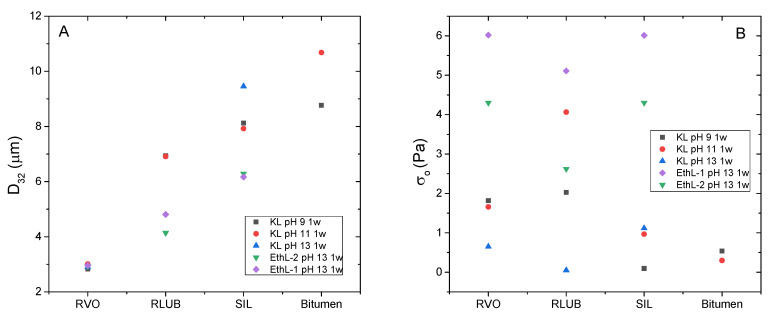
Sauter mean diameters (**A**) and yield stress values (**B**) of emulsions stabilized by bioethanol-derived and Kraft lignins formulated at different pH and stored for one week.

**Figure 10 polymers-13-02703-f010:**
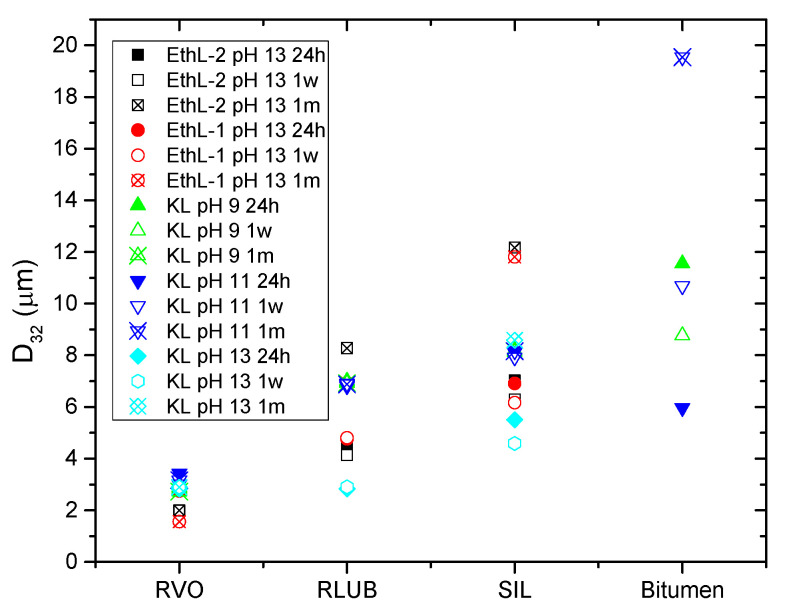
Evolution of emulsion droplet size with storage time as a function of the selected dispersed phase.

**Table 1 polymers-13-02703-t001:** Effect of oil concentration and lignin source on silicone emulsion viscous behaviour fitted to Herschel-Bulkley model. All emulsions were formulated at pH = 13 and measured after one week storage time.

**Viscous Model Parameters**	**EthL-1** **70% O**	**EthL-1** **65% O**	**EthL-1** **60% O**	**EthL-1** **50% O**	**EthL-1** **40% O**
σ_o_	10.05	6.02	2.4	0.90	0.56
K	25.16	0.30	0.13	0.10	0.07
n	0.35	0.79	0.82	0.64	0.58
**Viscous Model Parameters**	**EthL-2** **70% O**	**EthL-2** **65% O**	**EthL-2** **60% O**	**EthL-2** **50% O**	**EthL-2** **40% O**
σ_o_	4.65	4.30	2.70	0.77	0.5
K	14.9	0.33	0.17	0.04	0.04
n	0.31	0.74	0.80	0.81	0.75
**Viscous Model Parameters**	**KL** **70% O**	**KL** **65% O**	**KL** **60% O**	**KL** **50% O**	**KL** **40% O**
σ_o_	2.25	1.11	0.115	0.071	0.04
K	9.75	0.55	0.09	0.04	0.03
n	0.35	0.80	0.79	0.84	0.77

**Table 2 polymers-13-02703-t002:** Effect of formulation on emulsion storage stability, Z potential values (ζ) and Herschel-Bulkley model parameters. Measurements were performed after 7 days storage.

Oil	Surfactant	Oil wt.%	pH	Minimum Storage Stability (Days)	σ_o_ (Pa)	k (Pa s^n^)	n (−)	ζ (mV)
SIL	KL	65	9	30	0.09	0.07	0.89	–36.7
SIL	KL	65	11	30	1.11	0.36	0.88	–56.8
SIL	KL	65	13	30	1.11	0.55	0.80	–45.8
SIL	EthL-1	65	13	30	6.02	0.29	0.79	–42.2
SIL	EthL-2	65	13	30	4.30	0.32	0.74	–46.4
VGO	KL	65	9	30	1.82	0.33	0.78	–12.7
VGO	KL	65	11	30	1.66	0.15	0.93	–22.2
VGO	KL	65	13	30	0.65	0.25	0.73	–39.7
VGO	EthL-1	65	13	30	0.72	0.22	0.80	–46.0
VGO	EthL-2	65	13	30	0.86	0.33	0.74	–47.3
RLUB	KL	65	9	30	2.03	4.99	0.48	–6.8
RLUB	KL	65	11	30	4.06	3.92	0.52	–36.7
RLUB	KL	65	13	7	-	-	-	–42.2
RLUB	EthL-1	65	13	30	5.11	0.23	0.86	–36.5
RLUB	EthL-2	65	13	30	2.62	0.56	0.69	–42.7
Bitumen	KL	65	9	7	-	-	-	–29.8
Bitumen	KL	65	11	30	0.30	0.14	0.82	–28.1

## Data Availability

The raw/processed data required to reproduce these findings cannot be shared at this time due to technical or time limitations.

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
