# Peer review of "Bioethanol Production and Alkali Pulp Processes as Sources of Anionic Lignin Surfactants"

_polymers, 2021, doi:10.3390/polym13162703_

Round 1
Reviewer 1 Report
Congratulations to the authors for delivering a comprehensive and well organized study. Moreover, the extensive use of the literature to document the results is worth mentioning.
Author Response
Reviewer 1
Congratulations to the authors for delivering a comprehensive and well organized study. Moreover, the extensive use of the literature to document the results is worth mentioning.
Authors thank reviewer’s comments accepting this article in its current state.
Reviewer 2 Report
Comments:
- Describe in more detail the chemical structure and physicochemical properties of lignin, please.You can find the necessary information in articles by Pilarska et al., from recent years ( Energies (2018), Polymers (2019), Energies (2021)).
- In my opinion, this article should contain the summarized and structural formulas of the chemical compounds (selected, including lignins).
- Improve the quality and readability of Figure 1, please.
- Unify the font on all diagrams, I suggest using Arial.
- What are the practical benefits of the described method? What are the directions of implementation of this way of anionic lignin surfactants production? Which industries are likely to apply it?
- Complete the discussion on similar and competing sourcing methods of anionic lignin surfactants. Use the available literature data and include economic considerations, please.
Author Response
Reviewer 2
1. Describe in more detail the chemical structure and physicochemical properties of lignin, please.You can find the necessary information in articles by Pilarska et al., from recent years ( Energies (2018), Polymers (2019), Energies (2021)).
Physicochemical properties and chemical structure is mainly described in the Result and Discussion section, trying to correlated emulsification capability with the different lignin structure and functional groups expected to be present in bioethanol and Kraft lignins. However, additional description of lignin structure and functional groups have been addressed in the Introduction section using suggested references.
2. In my opinion, this article should contain the summarized and structural formulas of the chemical compounds (selected, including lignins).
Compounds used are mostly recycled and industrial products with a complex and mostly unknown composition, which cannot be easily summarized or described. Lignin compositions have been described in the experimental section according to information provided by suppliers. In addition, mention to the expected lignin structure have been included using references suggested in comment 1.
3. Improve the quality and readability of Figure 1, please.
Quality of Figure 1 has been improved and its content has been made clearer and readable.
4. Unify the font on all diagrams, I suggest using Arial.
All fonts have been unified using Arial
5. What are the practical benefits of the described method? What are the directions of implementation of this way of anionic lignin surfactants production? Which industries are likely to apply it?
Article proposes the use of anionic emulsions for asphalt (RAP) rejuvenators stabilized by native lignin as an alternative to cationic emulsions, previously designed by this authors, by amination of lignin to brings about a catatonic surfactant. Based on our previous experience, the non-polar character of the aged bitumen covering RAP may allow either the use of cationic or anionic emulsions. The use of a native lignin, instead of a modified one is the real benefit of the described method (please see reference 18 of this article). This point justifies the interest and novelty of this work and has been arisen in the fifth paragraph of the Introduction section. Interestingly, no implementation of the raw lignin production is needed (either for bioethanol process or kraft pulping). This article points out that both lignin sources may produce anionic surfactants able to stabilize asphalt rejuvenator, but results also show that alkaline pH must be carefully fixed according to the selected lignin-rejuvenator couple. Finally, this work focuses its application on the important road paving industry. However, other possible applications of such anionic surfactants could add value to this waste biopolymer products for cleaning, food, personal care, cosmetic, paint, biomedical, chemical, agricultural and pharmaceutical industry, waste purification and water treatment. This point has been addressed as another Concluding Remark.
6. Complete the discussion on similar and competing sourcing methods of anionic lignin surfactants. Use the available literature data and include economic considerations, please.
As reviewer suggests economic considerations, indicating that only a 2% of annual lignin production (estimated in 100 million tons) is used in commercial products, have been addressed in section 4-Concluding Remarks, in other to emphasize the interest in looking for value-added uses to this biopolymer. Similarly, alternative sources to obtain native lignin able to be used as anionic surfactants (e.g. sulphite, soda, organosolv processes, among others) have been addressed.